# Colorimetric Freshness Indicator Based on Cellulose Nanocrystal–Silver Nanoparticle Composite for Intelligent Food Packaging

**DOI:** 10.3390/polym14173695

**Published:** 2022-09-05

**Authors:** Seongyoung Kwon, Seonghyuk Ko

**Affiliations:** Department of Packaging, Yonsei University, Wonju 26493, Korea

**Keywords:** cellulose nanocrystal, silver nanoparticle, hydrothermal green synthesis, colorimetric freshness indicator, intelligent packaging

## Abstract

In this study, a colorimetric freshness indicator based on cellulose nanocrystal-silver nanoparticles (CNC-AgNPs) was successfully fabricated to offer a convenient approach for monitoring the quality of packaged food. AgNPs were directly synthesized and embedded in CNC via a one-pot hydrothermal green synthesis, and CNC-AgNP composited indicator films were prepared using a simple casting method. The AgNPs obtained were confirmed by UV-Vis diffuse reflectance spectroscopy and X-ray diffraction. The ability of the as-prepared CNC-AgNP film to indicate food quality was assessed in terms of the intensity of its color change when in contact with spoilage gases from chicken breast. The CNC-AgNP films initially exhibited a yellowish to dark wine-red color depending on the amount of AgNPs involved. They gradually turned colorless and subsequently to metallic grey. This transition is attributed to the reaction of AgNPs and hydrogen sulfide (H_2_S), which alters the surface plasmon resonance of AgNPs. Consequently, the color change was suitably discernible to the human eye, implying that the CNC-AgNP composite is a highly effective colorimetric freshness indicator. It can potentially serve as an accurate and irreversible food quality indicator in intelligent packaging during distribution or storage of products that emit hydrogen sulfide when deteriorating, such as poultry products or broccoli.

## 1. Introduction

Intelligent packaging is an advanced technique that can be used to monitor the external or internal environment of packaging and provide appropriate information to customers [1,2,3]. It is classified according to the type of smart device used, such as data carriers, sensors, and indicators. Barcodes and radio frequency identification tags (RFID) are typical data carriers that can provide information pertinent to supply chain management, facilitating inventory control and product identification [4]. Sensors typically include a receptor, transducer, signal-processing electronics, and signal display [5] and can be categorized as gas sensors or biosensors based on the type of receptor [6,7]. Time-temperature and shock-vibration indicators are representative indicators that are placed on the exterior of the packaging to record the surroundings. Other sensors include gas or freshness indicators placed within the packaging for the direct detection of gas, pathogens, pH, or their derivatives induced by food decay within the packaging. They primarily comprise redox dyes, pH dyes, or dyes that sense metabolites and convey information on food quality through color changes [1].

Freshness indicators based on gas or biosensors have been extensively studied because they can directly monitor the quality of packaged food. Recently, the replacement of typical dyes with metal nanoparticles (NPs) has attracted significant attention. Metal NPs change their color when a surface chemical reaction occurs due to specific chemicals that are released during food decay [8,9]. The highly sensitive and selective sensing characteristics of metal NPs, such as gold (Au) and silver (Ag), enable their application in gas detection [10,11,12,13] as well as biosensing and bioimaging [14,15].

Silver nanoparticles (AgNPs) are well-known nanomaterials used in sensing systems owing to their selective detection of hydrogen sulfide (H_2_S) and unique optical properties. When the diameter of the AgNP is considerably smaller than the wavelength of the incident light (R/λ << 0.1), the electron cloud on the AgNP oscillates owing to the attraction of protons and repulsion of the electromagnetic field [16]. If the oscillation is enhanced by incident light of a specific wavelength, the light scattering and absorption of AgNPs are maximized, resulting in a unique color expression [16,17]. This phenomenon, known as localized surface plasmon resonance (LSPR), is dependent on the density of the electric field located on the AgNPs, which can be varied by manipulating the size and shape of the AgNPs [18]. The surface chemical reaction between AgNPs and hydrogen sulfide (H_2_S), called sulfidation, induces a change in electron cloud density and intrinsic color of AgNPs. When AgNPs react with H_2_S under moist aerobic conditions at room temperature, the AgNP surface is transformed to silver sulfide (Ag_2_S) [19], resulting in a silver-silver sulfide (Ag@Ag_2_S) core-shell structure. Furthermore, the Ag_2_S layer changes the LSPR and refractive index of AgNPs in the visible range, thereby inducing a drastic color change in AgNPs [20].

H_2_S is the first derivative of the enzymatic degradation of sulfur-containing amino acids. Moreover, its odor can be sensed at low concentrations. Thus, it is considered an indicator of the freshness of meat [21]. Zhai et al. [12] studied AgNPs as an indicator to detect H_2_S and recorded food quality using the principle of the optical change exhibited by AgNPs. AgNPs were synthesized using gellan gum (GG) to form a composite, which was subsequently transformed into a hydrogel using agar. The GG-AgNP hydrogel selectively reacted with the H_2_S present in the spoilage gas from meat deterioration and transformed from yellow to colorless.

Cellulose nanocrystals (CNCs) obtained from natural resources such as wood, cotton, algae, tunicates, and bacteria have been widely applied in the food packaging industry. Especially, CNCs based on wood have remarkable surface chemical properties and mechanical strength compared to other cellulose-based materials, such as cellulose nanofibers (CNFs) and microcrystalline cellulose (MCC) [22,23,24,25]. In addition, the well-ordered cellulose molecules enable CNCs to form a rod-like structure with a high specific surface area. Furthermore, the numerous sulfate groups (OSO_3_^−^) on the CNC rods act as promising electron donors for the reduction of silver ions (Ag^+^) to silver (Ag^0^) [26].

The primary purpose of this study was to characterize and evaluate a freshness indicator that can detect H_2_S in chicken breast spoilage gas using a CNC-AgNP composite film prepared in a facile hydrothermal synthesis. The H_2_S detection properties and color changes of the CNC-AgNP films were investigated using a mechanism for the surface chemical reaction between AgNPs and H_2_S.

## 2. Materials and Methods

### 2.1. Materials

CNCs hydrolyzed by sulfuric acid were purchased as a freeze-dried powder from Celluforce™ (Montreal, QC, Canada). Sodium hydroxide (NaOH) was obtained from Sigma-Aldrich (St. Louis, MO, USA). Silver nitrate (AgNO_3_) was obtained from Alfa Aesar (Haverhill, MA, USA). H_2_S standard gas at a concentration of 2 ppm was purchased from RIGAS Co., Ltd. (Daejeon, Korea). The raw chicken breast used in this study was purchased from a local grocery store as slaughtered the day before. All materials were used without further purification.

### 2.2. Experiments

#### 2.2.1. Preparation of CNC-AgNP Solutions and Composite Films

CNC-AgNP solutions with various concentrations of AgNPs were synthesized using the following steps: the CNC powder was mixed with deionized water and vigorously stirred overnight. The CNC suspension was further dispersed using a probe sonicator (BCX 750, Sonics & Materials, Inc., Newtown, CT, USA) at 8000 J. NaOH solution (5 N) was added to the CNC suspension until a pH of 11 was attained. Various concentrations of AgNO_3_ solution were mixed with the CNC suspension. Subsequently, the hydrothermal synthesis of AgNPs was performed in an oven at 95 °C for 1 h. Finally, the CNC-AgNP solution was cooled overnight at room temperature.

CNC-AgNP films were prepared using a solution casting method [27]. Approximately 10 g of the as-prepared CNC-AgNP solution was poured into a disposable round polystyrene plate (60 mm in diameter and 15 mm in height) and dried in a desiccator for 7 days. Subsequently, the dried CNC-AgNP films were carefully retrieved from the plate and stored in a PET pouch. The nominal Ag content in the composite films was calculated using the weight ratio of Ag^+^ to CNC [28] and the samples were named accordingly (Table 1).

#### 2.2.2. Analysis of Sulfuric Compounds in the Spoilage Gas

Qualitative analysis of sulfur compounds was performed using a gas-chromatography-pulsed flame photometric detector (PFPD, Varian 450-GC, Bruker, Billerica, MA, USA). The spoilage gas in a 5 L Tedlar aluminum bag was concentrated at −15 °C using a Series 2 Unity equipped with an air server (Markes International Ltd., Llantrisant, UK), and inserted into the GC inlet at a flow rate of 15 mL/min of helium gas at 270 °C in split mode. CP-Sil 5CB (60 m × 0.32 mm inner diameter, 5 µm, Agilent J&W GC columns, Santa Clara, CA, USA) was used to separate sulfur compounds from the spoilage gas. The column was operated at a programming temperature of 80 °C for 3.5 min, increased at a rate of 6 °C/min to 150 °C, and maintained for 18 min.

To analyze H_2_S quantitatively, the spoilage gas was collected in a 250 mL glass jar with 5 g of raw chicken breast at 25 °C for 48 h. A gas sampling pump with an H_2_S gas detector tube extracted 100 mL of headspace gas through a septum on the glass jar cap. Gases were sampled six times for a predetermined storage time, and three samples were considered each time to obtain the final average value. The detector tubes used (GASTEC, Seoul, Korea) were 4LT (0.05–4.0 ppm), 4LB (0.5–12 ppm), and 4LK (1–40 ppm).

#### 2.2.3. Characterization of CNC-AgNP Solutions and Composite Films

The light absorption properties of the samples were measured using a UV-Vis spectrophotometer (V-650, Jasco, Tokyo, Japan) in the wavelength range 300–800 nm. The surfaces of the neat CNC and CNC-AgNP films were observed with field emission scanning electron microscopy (FE-SEM, Quanta FEG 250, FEI Co., Ltd., Hillsboro, OR, USA) at magnifications up to 8000×. X-ray diffraction (XRD, D2 Phaser model system, Bruker, Billerica, MA, USA) with Cu Kα radiation (λ = 1.5418°) was used to identify the crystallinity of the films. Finally, XRD patterns were obtained in the 2θ range of 5–80°.

#### 2.2.4. Evaluation of CNC-AgNP Composite Films as a Colorimetric Freshness Indicator

The H_2_S sensing test was conducted with the C-0.22Ag film in contact with H_2_S standard gas. The film was placed under the cap of a 250 mL glass jar with 30 mL of water. The headspace was evacuated of gases over a period of 1 min and then filled with H_2_S standard gas over a period of 25 min at a flow rate of 10 mL/min. The tested film was analyzed using a UV-Vis spectrophotometer, and its color change was observed as a function of contact time.

The ability of the film to detect spoilage gas from raw chicken breast decomposition was evaluated. The film sample (3 cm × 4 cm) was installed in the same glass jar with 5 g of raw chicken breast and was maintained at 25 °C for 48 h. In addition, photographs of the change in film color with storage time were taken. The light absorption properties of the films were measured using a UV-Vis spectrophotometer. Finally, the changes in the morphology and crystallinity of the films were investigated using FE-SEM and XRD.

## 3. Results and Discussion

### 3.1. Confirmation of AgNPs Formation

The appearances of the as-prepared CNC and CNC-AgNP composite samples are presented in Figure 1. A distinct color was observed when AgNO_3_ was added to the CNC suspension, resulting in an orange or dark brown color. From the UV-Vis spectra of the samples illustrated in Figure 2, both the CNC-AgNP solutions and composite films displayed strong absorption peaks at approximately 418 nm, which were not detected in the CNC suspension and film. Previous studies have reported that silver nanospheres with a diameter of approximately 40 nm absorb light of wavelength 410 nm and exhibit a unique color [17,29]. The CNC-AgNP samples exhibited similar light absorption properties and colors, indicating the presence of spherical AgNPs.

Figure 3 depicts the XRD patterns of the neat CNC and CNC-AgNP composite films. Two remarkable peaks at 2θ = 15.4° and 22.7°, corresponding to CNC(101) and CNC(002) [22], were observed in all films. New peaks at approximately 2θ = 38.1° and 44.3° were detected in the CNC-AgNP composite films, corresponding to Ag(111) and Ag(200), respectively [30]. Thus, the conclusion can be drawn that AgNPs were successfully synthesized through green hydrothermal synthesis using CNCs.

### 3.2. Sulfur Compounds in Spoilage Gas

The qualitative analysis of sulfur compounds from the deterioration of raw chicken breast revealed that six different sulfuric chemicals were produced: hydrogen sulfide (H_2_S), methanethiol (CH_3_SH), carbon disulfide (CS_2_), dimethyl sulfide ((CH_3_)_2_S), dimethyl disulfide ((CH_3_)_2_S_2_), and dimethyl trisulfide ((CH_3_)_2_S_3_). The presence of these chemicals can be attributed to the high protein content of raw chicken breast [31]. Previous studies have reported that the enzymatic degradation of cysteine, which is a sulfur-containing amino acid in proteins, generates diverse sulfides, from low molecular sulfides to polysulfides [21].

Figure 4 shows the H_2_S generation tendency against various storage times at room temperature using a gas detector tube. H_2_S was not detected in the initial 8 h, after which its concentration progressively increased up to 32.55 ppm. Typically, H_2_S is generated from the degradation of meat products by microbes under anaerobic conditions and the depletion of glucose [32]. Sukhavattanakul and Manuspiya reported that minced pork stored in a closed vial at 4 °C produced H_2_S, which was first detected after eight days and eventually peaked at a concentration of 250 ppm [33]. In this study, the transition at 8 h can be attributed to the fact that suitable conditions for microbial decomposition were formed at this point, resulting in an accelerated increase in H_2_S concentration.

### 3.3. Reaction Property of CNC-AgNP Films to H_2_S Standard Gas

The UV-Vis spectra and photographs of the C-0.22Ag film after contact with H_2_S standard gas are presented in Figure 5 and Figure 6, respectively. The intensity of color change peaked at approximately 430 nm; it started to decline at 5 h and completely disappeared after 30 h of reaction. The change in color was first observed at 5 h; the film gradually turned dark brown and eventually disintegrated. This phenomenon can be attributed to the formation of Ag_2_S, which causes the LSPR of AgNPs to change. Park et al. reported that the surface chemical reaction between AgNPs and H_2_S results in the formation of the Ag@Ag_2_S core-shell structure. Consequently, the diameter, LSPR, and refractive index of AgNPs in the visible region change [20].

As illustrated in Figure 7, Lilienfeld and White suggested that the sulfidation of AgNPs was induced by H_2_S [19]. When H_2_S molecules contact the AgNP surface under moisturized conditions, they are oxidized to water and sulfur by the oxygen molecules on the AgNPs. The oxidation of H_2_S immediately causes the Ag and sulfur atoms to combine and form Ag_2_S. The sulfidation process can be summarized as follows:(1)H2S+12O2 → H2O +S
(2)$$S+2\mathrm{Ag}\;\to {\;\mathrm{Ag}}_{2}S$$
(3)2Ag+ H2S+12O2 → Ag2S +H2O under moisture condition

### 3.4. Quality Indicating Performance of CNC-AgNP Film

Figure 8 shows the color change of the neat CNC and CNC-AgNP films after contact with the chicken breast spoilage gas. The initial color remained unchanged up to 8 h, following which a weak purple color appeared after 16 h in the case of the neat CNC film. In addition, studies have reported a color change in the visible-light range if moisture is absorbed in the chiral nematic structure of CNC rods [34]. Thus, the color change of the neat CNC film can be attributed to the presence of moisture, and not H_2_S. However, no distinct color change was observed for any of the CNC-AgNP films up to this stage because no H_2_S had been generated, as shown in Figure 4. Color change in the C-0.22Ag film was first detected at 8 h into the experiment. The color change from reddish-brown to grayish yellow was completed at 24 h. However, the films containing over 1.12 wt.% AgNP behaved differently compared to the C-0.22Ag film. The C-1.12Ag film brightened progressively by a marginal amount until 16 h, and a grayish stain was observed after 24 h. In the case of the C-2.25Ag and C-4.49Ag films, the color intensity decreased progressively until 8 h. Metal-like surfaces were observed in both films at 16 h, and a glossy surface was observed in the C-4.49Ag film after 24 h. As shown in Figure 8, all the CNC-AgNP composite films reacted sensitively with H_2_S and changed color. However, the C-0.22Ag film is the most promising for practical application owing to its higher Ag content, which results in a larger metallic surface.

Figure 9 displays the surface morphologies of the neat CNC and CNC-AgNP films after reacting with H_2_S for 48 h. The surface of the C-0.22Ag film transformed into a dotted pattern, which was observed at a magnification of 8000×. When the weight ratio of AgNP was increased, crack formation was initiated and a white spot-like pattern began to appear on the C-1.12Ag film surface after reaction with the spoilage gas. This tendency was intensified in the C-2.25Ag film, which exhibited larger crack patterns and white markings. In the case of the highest concentration of AgNPs, white particles were observed on the severely damaged surface. In the XRD patterns displayed in Figure 10, two peaks at 2θ = 15.4° and 22.7°, corresponding to CNCs [22], were intact after contact with the spoilage gas. When the CNC-AgNP films were exposed to the spoilage gas, several peaks at 2θ = 28.9°, 31.8°, 33.6°, 34.4°, 36.8°, 38.1°, 40.6°, and 52.7° matched to Ag_2_S [35] were observed. These peaks were intensified and readily discerned when the Ag content in the CNC-AgNP films was increased.

The FE-SEM and XRD analyses revealed that H_2_S did not affect the crystallinity of CNCs, but only reacted with AgNPs, resulting in the formation of Ag_2_S. Ma et al. [36] presented a similar result: Ag_2_S appeared as a small white particle on the film. In this study, Ag_2_S particles appeared in a dotted pattern on the C-0.22Ag film after the reaction, as shown in Figure 9. It evolved into crack-patterned and damaged surfaces in the case of films with Ag content over 1.12 wt.%. This phenomenon was observed simultaneously with an increase in the intensity of the XRD patterns assigned to Ag_2_S, which implies that a larger amount of Ag_2_S particles was initially formed. The particles, subsequently, aggregated owing to the high concentration of AgNPs [37] and eventually covered the film surface. The severely damaged surface observed in the tested C-4.49Ag film surface was a result of chemical corrosion caused by the high Ag_2_S content [38]. Therefore, the conclusion was drawn that sulfidation of AgNPs was induced by the H_2_S in the spoilage gas from the decaying raw chicken breast, which was dependent on the Ag content of the CNC-AgNP films.

## 4. Conclusions

We successfully prepared CNC-AgNP nanocomposites that can be used as colorimetric freshness indicators of packaged food, owing to the LSPR of AgNPs. The presence of AgNPs synthesized within the CNCs was confirmed using UV-Vis spectroscopy and XRD patterns. The ability of the CNC-AgNP films to indicate food quality was evaluated in terms of how effectively they detected spoilage gas emitted from decomposing raw chicken breast. In addition, UV-Vis, FE-SEM, and XRD were used to characterize this ability. Prior to testing, spoilage gases were analyzed specifically for sulfuric compounds, and the generation of H_2_S during raw chicken breast decay was confirmed using GC-PFPD. The CNC-AgNP films exhibited an exceptional ability in detecting the spoilage gases, with a noticeable color change from yellowish-dark wine red to colorless metallic silver depending on the AgNP content. Furthermore, the results revealed that the sulfidation of AgNPs by H_2_S induced LSPR changes in the AgNPs under moisturized and low-oxygen conditions, resulting in a color change in the CNC-AgNP nanocomposite film. Finally, the XRD results confirmed that Ag_2_S was formed, and SEM revealed that the surface of films with a higher Ag content was damaged by chemical corrosion caused by Ag_2_S.

## Figures and Tables

**Figure 1 polymers-14-03695-f001:**
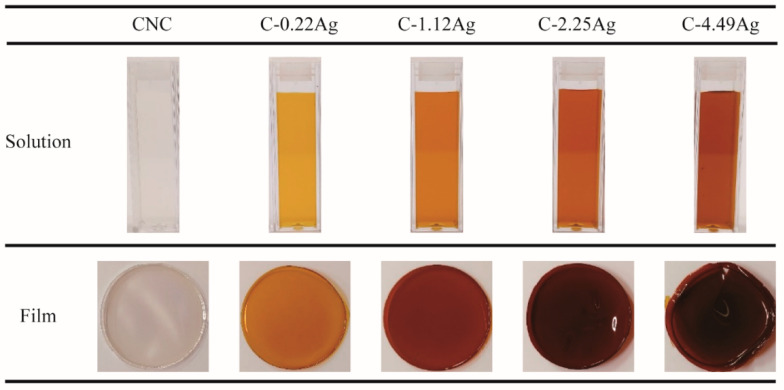
Photographs of CNC and CNC-AgNP composite samples.

**Figure 2 polymers-14-03695-f002:**
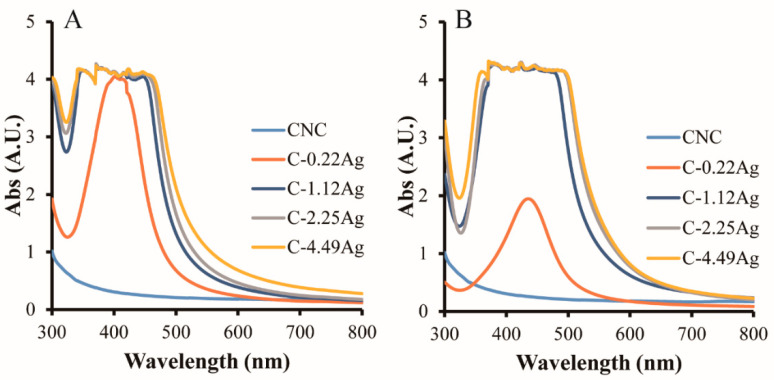
UV-Vis spectra of neat CNC and CNC-AgNP composites; liquids (**A**) and composite films (**B**).

**Figure 3 polymers-14-03695-f003:**
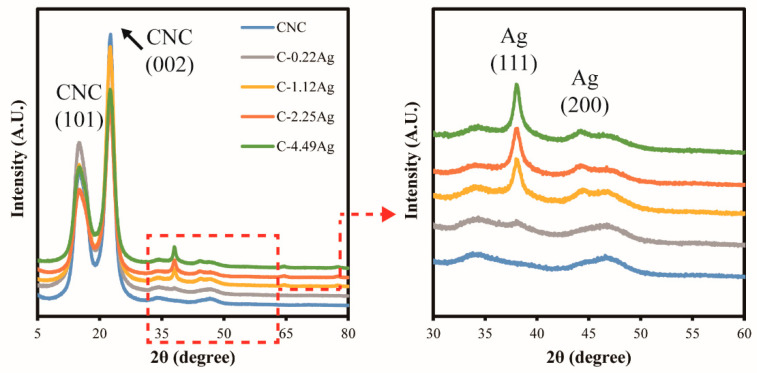
XRD patterns of neat CNC and CNC-AgNP composite films.

**Figure 4 polymers-14-03695-f004:**
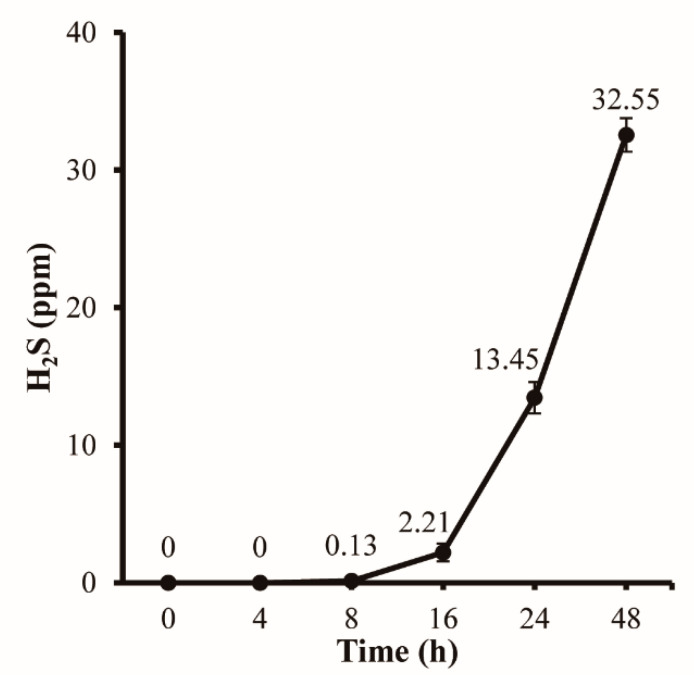
H_2_S concentration along with storage time at room temperature.

**Figure 5 polymers-14-03695-f005:**
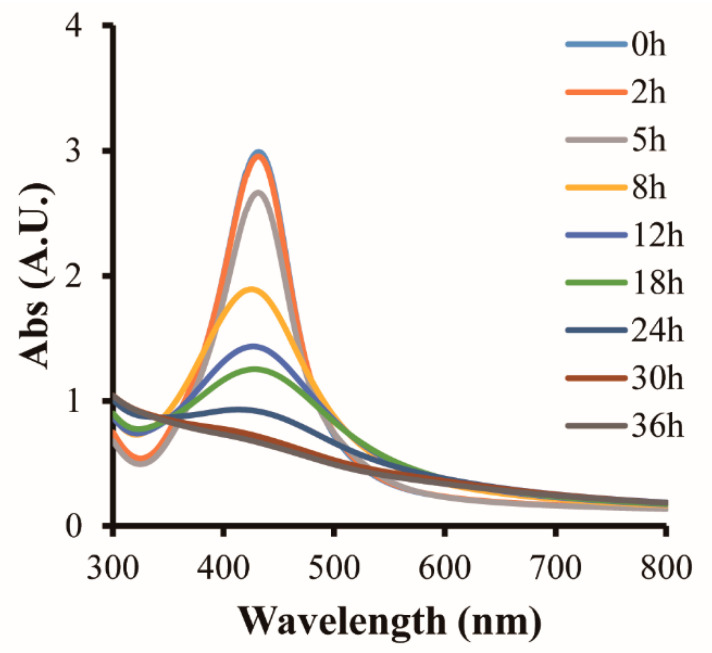
UV-Vis spectra of C-0.22Ag film after reaction with 2 ppm H_2_S.

**Figure 6 polymers-14-03695-f006:**
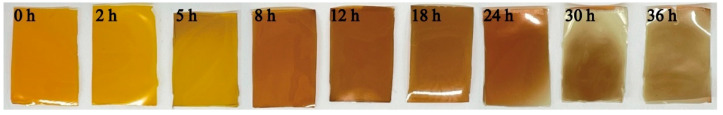
Photographs of C-0.22Ag film after reaction with 2 ppm H_2_S.

**Figure 7 polymers-14-03695-f007:**
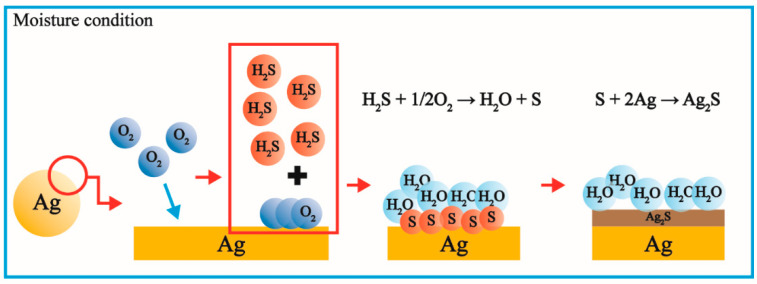
Schematic diagram of sulfidation of AgNP.

**Figure 8 polymers-14-03695-f008:**
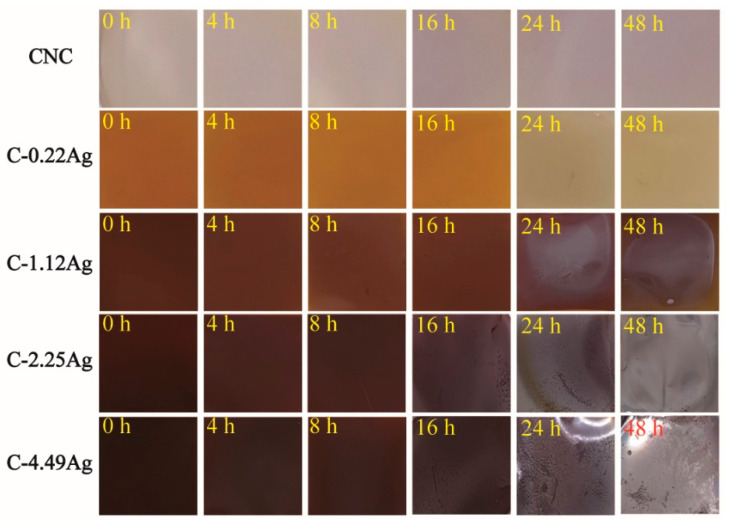
Photographs of neat CNC and CNC-AgNP films after contact with the chicken breast spoilage gas.

**Figure 9 polymers-14-03695-f009:**
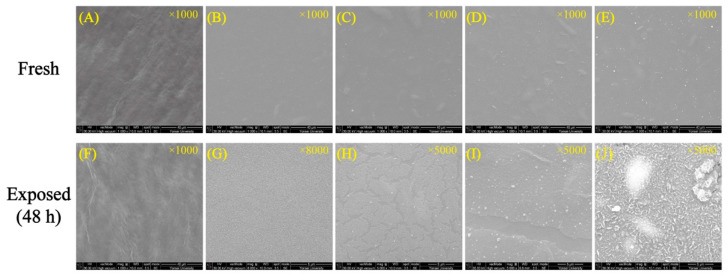
FE-SEM images of fresh (**A**–**E**) and exposed films to spoilage gas (**F**–**J**); neat CNC (**A**,**F**), C-0.22Ag (**B**,**G**), C-1.12Ag (**C**,**H**), C-2.25Ag (**D**,**I**), and C-4.49Ag films (**E**,**J**).

**Figure 10 polymers-14-03695-f010:**
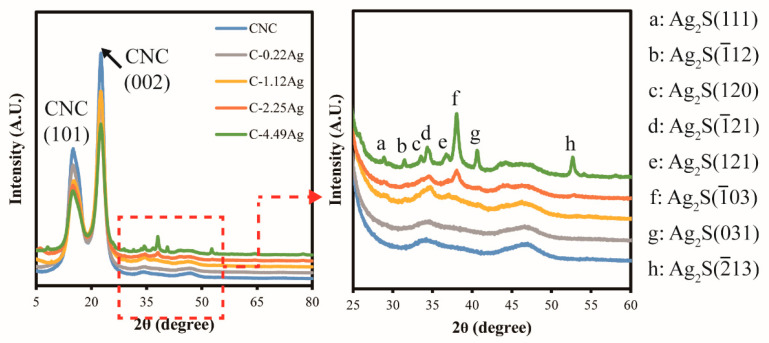
XRD patterns of tested CNC and CNC-AgNP films.

**Table 1 polymers-14-03695-t001:** Nominal weight ratio of AgNPs in CNC-AgNP solution.

2 wt.% CNC Suspension (g)			Water (g)	CNC (g)	AgNO_3_ (g)	CNC-AgNP Solution (g)	
48			47.04	0.96	2	50	
**Conc. (mM)**	**AgNO_3_ (g)**	**Mol**	**Mass (g)**	**Ag^+^ mass (g)**	**CNC-AgNP ratio in CNC-AgNP solution (wt.%)**		**Sample code**
CNC only							CNC
10	2	2.0 × 10^−5^	3.40 × 10^−3^	2.16 × 10^−3^	0.22		C-0.22Ag
50		1.0 × 10^−4^	1.70 × 10^−2^	1.08 × 10^−2^	1.12		C-1.12Ag
100		2.0 × 10^−4^	3.40 × 10^−2^	2.16 × 10^−2^	2.25		C-2.25Ag
200		4.0 × 10^−4^	6.79 × 10^−2^	4.31 × 10^−2^	4.49		C-4.49Ag

## Data Availability

The original contributions presented in the study are included in the article, further inquiries can be directed to the corresponding author.

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
