# Peer review of "Colorimetric Freshness Indicator Based on Cellulose Nanocrystal–Silver Nanoparticle Composite for Intelligent Food Packaging"

_polymers, 2022, doi:10.3390/polym14173695_

Round 1

Reviewer 1 Report

The study established a method of checking H2S based on cellulose nanocrystal-silver nanoparticles to monitoring the quality of packaged food. Generally, the research has certain reference value and significance, and the research workload is moderate.

1        Writing standard: 1) Line 101and 103,“petri dish”, non-conformant .2)  Line 188 -192, discuss the same reference? if it is a document, please cited it in one sentence; if it is not, removed “.”in line 189, become a sentence. And so on.

2        There are two main mechanisms for conversion of AgNPs to vulcanized AgNPs: (1) When AgNPs is oxidized and dissolved under aerobic conditions, silver ions react with sulfur in the environment to indirectly form silver sulfide; (2) In anaerobic environment, when sulfur ion concentration is high enough, negative divalent sulfur ions can directly convert AgNPs into silver sulfide.  Figure 6 showed the results under aerobic conditions, In addition to generating Ag2S, whether there was Ag2S? For packaged foods, most of them should be oxygen-free, the schematic wasn't unbalanced.

3        Experimental material, raw chicken breast. Purchased or slaughterd immediately after the experimental study? If purchased, then how long after slaughter? After 8 hours as shown in Figure 4,  the original condition of raw chicken  breast should be clearly indicated

4         Reasons for choosing C-0.22Ag Film,as XRD patterns shown  C-4.49Ag film With the most obvious peaks and steady state。

5        Figure 4 shown the concentration of the gas increases over time. And Figure 5 shown the color was constantly changing of C-0.22Ag film after reaction with 2 ppm H2S over time.

6        So how to determine the concentration of H2S, because C-Ag film changingcolor as time goes on at a constant concentration of H2S , Or FOR  the food(raw chicken breast ) produces more H2S over time? And on the basis of time variation, limit of detection?

Reviewer 2 Report

The study is generally very well-designed I have only a few suggestions and questions.

In the current study, the authors create a colorimetric freshness indicator based on cellulose nanocrystal-silver nanoparticles (CNC-AgNPs). I wonder if bacterial cellulose could be used in a similar way. Recently, this material has become very attractive in the food industry. The authors could consider supplementing the information about bacterial cellulose and the possibility of its use as food packaging.

In line 71-73 you write that "Cellulose nanocrystals (CNC) obtained from wood by sulfuric acid hydrolysis". Next in lines 78-79 you write that your CNC-AgNP composite films was prepared in an eco-friendly manner. I do not fully agree with this statement. Please use a different phrase or explain in more detail in the introduction why this composite is environmentally friendly.

The qualitative analysis of sulfur compounds from the deterioration of raw chicken breast revealed that six different sulfuric chemicals were produced. In the case of meat other than poultry, would the gas composition and the speed of their production be similar?Are your composites universal for meat from different animals?

Reviewer 3 Report

1. Were the AgNPs incorporated into or conjugated with the CNC?

2. How about the water sensibility of the resulted film? Since the film should be applied in food products, have the authors considered how it is going to be used? Wrapped or coated?

3. Figure 5 should be divided into 2 parts, UV-VIS spectra and photographs.

Round 2

Reviewer 1 Report

 The authors have better revised and answered the questions, and made some modifications.

Reviewer 3 Report

I suggest this study can be accepted now.

Author Response

I suggest this study can be accepted now.

(Answer) The authors are grateful for this comment.